# A Survey of Methodologies for Assessing Mast Cell Density and Activation in Patients with Functional Abdominal Pain Disorders

Hunter Friesen [1], Meenal Singh [2], Vivekanand Singh [3], Jennifer V. Schurman [2,4] and Craig A. Friesen [2,4,*]

1   School of Medicine, University of Kansas, Kansas City, KS 66160, USA; hfriesen2424@gmail.com
2   Division of Gastroenterology, Hepatology, & Nutrition, Children's Mercy Kansas City, 2401 Gillham Road, Kansas City, MO 64108, USA; msingh1@cmh.edu (M.S.); jschurman@cmh.edu (J.V.S.)
3   Department of Pathology, The University of Texas Southwestern Medical Center, 1935 Medical District Drive, Dallas, TX 75235, USA; Vivekanand.Singh@UTSouthwestern.edu
4   Department of Pediatrics, School of Medicine, University of Missouri-Kansas City, Kansas City, MO 64108, USA
*   Correspondence: cfriesen@cmh.edu

**Abstract:** The aim was to assess methods utilized in assessing mast cell involvement in functional abdominal pain disorders (FAPDs), specifically to describe variability in methods utilized to assess both mast cell density and activation and determine if a consensus exists. After a literature search identified 70 manuscripts assessing mast cell density, data were extracted including FAPD diagnosis, site of biopsy, selection of microscopic fields analyzed, selection of mucosal region analyzed, method of mast cell identification, method to assess mast cell density, and if performed, method to assess mast cell activation. There appears to be some consensus favoring inmmunohistochemical stains over histochemical stains for identifying mast cells. Otherwise, considerable variability exists in methodology for assessing mast cell density and activation. Regardless of method, approximately 80% of studies found increased mast cell density and/or activation in comparison to controls with no method being superior. A wide variety of methods have been employed to assess mast cell density and activation with no well-established consensus and inadequate data to recommend specific approaches. The current methodology providing physiologic information needs to be translated to a standard methodology providing clinical information with the development of criteria establishing abnormal density and/or activation, and more importantly, predicting treatment response.

**Keywords:** mast cells; irritable bowel syndrome; functional dyspepsia

## 1. Introduction

Functional abdominal pain disorders (FAPDs), particularly irritable bowel syndrome (IBS) and functional dyspepsia (FD), are highly prevalent conditions resulting in significant morbidity and healthcare costs worldwide. IBS is defined by abdominal pain associated with a change in stool frequency, a change in stool form, or a change in pain intensity with stools [1]. FD is defined by the presence of epigastric pain, epigastric burning, early satiety, or postprandial fullness [2]. FAPDs are considered to result from disordered brain-gut axis function with a wide variety of pathophysiologic contributors including altered neural pathways (both peripheral and central), dysmotility, visceral hypersensitivity, inflammation, dysbiosis, and psychologic dysfunction. Mast cells have been implicated in both IBS and FD, in part due to their location at the interface between the patient and the environment and in part due to their functional connectivity to the multiple systems implicated in the generation of gastrointestinal symptoms [3]. Previous studies have shown increased mast cell density and/or evidence of increased mast cell activation in patients with FAPDs in most, but not all, studies. A meta-analysis of adults with FD demonstrated increased mast cells in the stomach and duodenum [4]. Multiple systematic reviews and/or

meta-analyses have demonstrated increased mast cells in the ileum and colon of adults with IBS [5–8]. In the largest of these reviews, Krammer and colleagues reviewed 36 studies with 30 of these studies demonstrating increased mucosal mast cells in adults with IBS [8]. Mast cells have been specifically linked to the development of visceral hypersensitivity, a pathophysiologic process of central importance in FAPDs [9].

A variety of methods have been employed to assess gastrointestinal mast cells, most commonly methods to determine mast cell density. There are a variety of techniques utilized to identify mast cells including histochemical staining (e.g., utilizing toluidine blue or Alcian blue) or immunohistochemical staining (e.g., utilizing antibodies to tryptase or CD 117, also known as c-kit). However, it is well recognized that mast cells exert their biologic functions primarily through the release of mediators and, thus, density does not give a complete picture of mast cell involvement. A variety of methods are available to assess mast cell activation. These include measuring (1) degranulation (e.g., utilizing transmission electron microscopy) [10–19]; (2) mast cell-derived mediators (e.g., tryptase and histamine) in biologic fluids or intestinal tissue either utilizing techniques such as RNA seq or protein analysis [17,18,20–23]; and, (3) mediators in the supernatant after tissue incubation [16,17,20,24–33].

The aim of the current survey was to assess the methods utilized in assessing mast cell involvement in FAPDs in both adults and children, specifically to describe variability in methods utilized to assess both mast cell density and activation and determine if there appears to be any consensus. Ultimately, the transition of the current state implicating mast cells in the pathophysiology of FAPDs to a clinically useful process of assessing mast cell involvement will require some standardization of methods, definitions of abnormal, and proof of an ability to predict response to treatments directed at mast cells or their mediators. Assessment of current practices represents the first step in this transition.

## 2. Literature Assessment

We conducted a literature search utilizing PubMed, Google, and Google Scholar employing the keywords "gastrointestinal mast cells", "irritable bowel syndrome", and "functional dyspepsia" from 2000 to 2021. Additionally, we cross-referenced bibliographies from all identified manuscripts to identify other relevant manuscripts. While the current manuscript was intended to be a methodologic survey and not a systematic review, we did include all relevant manuscripts identified in previous systematic reviews assessing mast cells in patients with FD and/or IBS [4–8]. Manuscripts were included if they reported on any patients with FAPDs and included an assessment of gastrointestinal mast cell density. Manuscripts were excluded if they did not assess density, even if activation was assessed.

Data extractions were performed independently by 2 reviewers (HF and CF). After comparing results, any discrepancies were resolved. Specific data extracted was first author, country where subjects were evaluated, age group, patient FAPD diagnosis, site of biopsy, selection of microscopic fields analyzed, selection of mucosal region analyzed, method of mast cell identification, method to assess mast cell density, and if performed, method to assess mast cell activation (See Figure 1). In addition, whether density and activation differed from a control group was assessed, primarily to determine how frequently findings related to density were discrepant from findings related to activation.

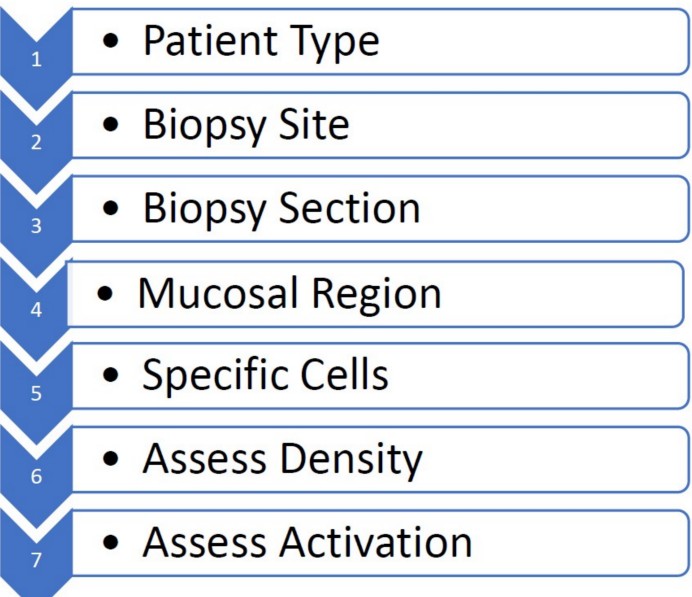

**Figure 1.** Levels of study design decisions regarding assessment of mast cells in functional abdominal pain disorders.

### 3. Summary of Methods for Mast Cell Evaluation

A total of 11 studies (1 IBS, 5 FD, 2 FAPD, and 3 of endoscopy patients) in children and adolescents and 59 studies (44 IBS, 14 FD, and 1 with both IBS and FD) in adults were identified [8–78]. The primary findings in pediatric patients are shown in Table 1 and in adults are shown in Tables 2 and 3. In youth, 8/11 (73%) studies identified mast cells with immunohistochemical stains (all utilizing anti-tryptase antibody) and 3/11 studies utilized histochemical stains. In adults, 49/59 (83%) studies identified mast cells with immunohistochemical stains (33 utilizing anti-tryptase, 15 anti-CD117, and one using both) and 10/59 studies utilized histochemical stains, most commonly toluidine blue (7 studies). Mast cell activation was assessed in 2/11 (18%) pediatric studies and 25/59 (43%) of adult studies. The number of microscopic fields assessed varied from 5 to 10 in pediatric studies and 3/20 in adult studies. In pediatric studies, the process for field selection was not stated in 3 studies, random in 2 studies, and through identification of most involved areas in 6 studies. In adult studies, the process for field selection was not stated in 42 studies, random in 11 studies, through identification of most involved areas in 2 studies, and through identification of 'most representative' areas in 4 studies. Other selection criteria in some studies included a requirement that the specimen be well-oriented or that villi be cut transversely. Others also avoided lymphoid follicles or lymphoid aggregates. Additional variability was noted in the mucosal layer evaluated with some assessing only the lamina propria and some assessing both the lamina propria and epithelium. There were also a variety of methods for assessing the mast cell density. While most studies involved manual counting of mast cells, others utilized image analysis to report the percentage of the area occupied by mast cells (either overall or lamina propria only). With manual counting, mast cell density was reported per high power field in some studies, per $mm^2$ in others, and as a percentage of total immunocytes in another. Densities were also reported by both quantitative and semi-quantitative (density ranges) methods.

**Table 1.** Summary of pediatric studies assessing mast cell density in functional abdominal pain disorders.

| Author | Country | Age Group | Population (N) | Mucosal Sites | Mast Cell ID Method | Number of Microscopic Fields Assessed | Field Selection | Cell Activation Assessed |
|---|---|---|---|---|---|---|---|---|
| Yeom et al. [34] | Korea | 6–12 | FD (56) | Gastric antrum and body; duodenum | Anti-tryptase | 5 | Most involved area | No |
| Henderson et al. [35] | USA | 5–17 | AP-FGID (26) | Upper and lower | Toluidine Blue | 10 | Random | No |
| Di Nardo et al. [36] | Italy | 4–18 | IBS (21) | TI, Ascending and Descending Colon | Anti-tryptase | Not stated | Random | No |
| Mahjoub et al. [37] | Iran | 1–14 | Endoscopy patients (86) | Antrum | Giemsa | 10 | Not stated | No |
| Schurman et al. [38] | USA | 8–17 | FD (59) | Antrum and duodenum | Anti-tryptase | 5 | Most involved area | No |
| Singh et al. [39] | USA | 8–17 | FD (114) | Antrum and duodenum | Anti-tryptase | 5 | Most involved area | No |
| Schäppi et al. [40] | UK | 2–12 | FD (16) | Gastric | Anti-tryptase | 10 | Not stated | Yes |
| Saad et al. [41] | USA | 3.3–17.9 | Endoscopy patients: 92% for abdominal pain (41) | Cecum, ascending, transverse, descending and rectosigmoid colon | Anti-tryptase | 5 | Most involved area | No |
| Friesen et al. [42] | USA | 8–17 | FD (30) | Antrum | Anti-tryptase | 5–10 | Not stated | Yes |
| Chernetsova et al. [43] | Canada | 1–17 | Endoscopy patients designated as healthy (38) | Gastric body and antrum, duodenum, TI, cecum, ascending, transverse, descending, and sigmoid colon, and rectum | Hematoxylin-Phloxine-Saffron and Giemsa | Not stated | Most involved area | No |
| Friesen et al. [44] | USA | 8–17 | AP-FGID (208) | Antrum and duodenum | Anti-tryptase | 5 | Most involved area | No |

**Table 2.** Summary of adult studies assessing mast cell density where mast cell activation was not assessed in functional abdominal pain disorders.

| Author | Country | Age Group | Population (N) | Mucosal Sites | Mast Cells ID Method | Number of Microscopic Fields Assessed | Field Selection | Density Different from Controls |
|---|---|---|---|---|---|---|---|---|
| Goral et al. [45] | Turkey | Mean 35–36 years | IBS (72) | Cecum and rectum | Giemsa | 10 | Not stated | Yes |
| De Silva et al. [46] | Sri Lanka | 18–59 years | IBS-D (49) | Ileum, cecum, ascending, transverse, descending, and rectum | Giemsa | 10 | Not stated | Yes |
| Binesh et al. [47] | Iran | 15–76 years | FD (25) | Stomach and duodenum | Giemsa | ≥5 | Not stated | No |
| Tunc et al. [48] | Turkey | 27–64 years | IBS (11) | Cecum | Toluidine blue | 10 | Not stated | Yes |
| Chadwick et al. [49] | New Zealand | 19–79 years | IBS (77) | Ascending, transverse, descending, and rectum | Tryptase | 15 | Not stated | Yes |
| Wang et al. [50] | China | Mean 42–49 years | IBS-D (20) and IBS-C (18) | Duodenum, jejunum, and TI | Tryptase | 6 | Not stated | Yes |
| Yang et al. [51] | China | 16–75 years | IBS-D (55) | TI, ascending and sigmoid | Tryptase | Not stated | Not stated | Yes |
| Chang et al. [52] | USA | 18–55 years | IBS-PI (45) | Sigmoid | Tryptase | % of area | Not stated | No |
| Sohn et al. [53] | Korea | 18–72 years | IBS-D (22) | Rectum | Tryptase | Not stated | Most representative | Yes |
| Ahn et al. [54] | Korea | Median 32 years | IBS-D (83) | Ascending, transverse, descending, sigmoid, and rectum | Tryptase | 6 | Not stated | Yes |
| Dunlop et al. [55] | UK | Mean 38–40 years | IBS (75) | Rectum | Tryptase | 4 | Not stated | Yes |
| El-Sahly et al. [56] | Norway | 18–62 years | IBS (50) | Rectum | Tryptase | 10 | Random | No |
| Cremon et al. [57] | Italy | 22–75 years | IBS (48) | Descending colon | Tryptase | % of area | Random | Yes |
| Dunlop et al. [58] | England | Mean 42 years | IBS-PI (28) | Rectum | Tryptase | 4 | Not stated | No |
| Bian et al. [59] | China | 21–66 years | D-IBS (10) | Descending colon | Tryptase | ≥10 | Random | Yes |
| Sundin et al. [60] | Sweden | Mean 32 years | IBS (43) | Sigmoid colon | Tryptase | 3 | Not stated | No |

**Table 2.** *Cont.*

| Author | Country | Age Group | Population (N) | Mucosal Sites | Mast Cells ID Method | Number of Microscopic Fields Assessed | Field Selection | Density Different from Controls |
|---|---|---|---|---|---|---|---|---|
| O'Sullivan et al. [61] | Ireland | 28–65 years | IBS (14) | Cecum, ascending, descending, and rectum | Tryptase | 3 | Not stated | Yes |
| Park et al. [62] | Korea | 25–65 years | IBS-D (18) | TI, ascending, and rectum | Tryptase | 6 | Not stated | Yes |
| Lee et al. [63] | South Korea | Mean 48 years | IBS (42) | Rectum | Tryptase | 5 | Not stated | Yes |
| Kim et al. [64] | Korea | Mean 30–51 years | IBS (18) | Descending, sigmoid, and rectum | Tryptase | 5 | Not stated | Yes |
| Giancola et al. [65] | Belgium | 18–68 years | FD (13) | Duodenum | Tryptase | 4 | Random | Yes |
| Hall et al. [66] | Ireland | 18–79 years | FD (62) | Gastric body and antrum | Tryptase | 15 | Not stated | Yes |
| Vanheel et al. [67] | Belgium | 17–52 years | FD (15) | Duodenum | Tryptase | $\geq 7$ | Representative | Yes |
| Tanaka et al. [68] | Japan | Mean 45 years | FD (9) | Duodenum | Tryptase | 5 | Not stated | No |
| Vicario et al. [69] | Spain | 18–63 years | IBS-D (49) | Jejunum | CD117 | 8 | Not stated | Yes |
| Braak et al. [70] | Amsterdam | 19–65 years | IBS (66) | Ascending and descending colon | CD117 | 18 | Not stated | Yes- decreased |
| Boyer et al. [71] | France | Mean 54–67 years | IBS (11) | Cecum, transverse, descending, and rectum | CD117 | 4 | Not stated | Not reported |
| Piche et al. [72] | France | Mean 54 years | IBS (50) | Cecum | CD117 | 5 | Not stated | Yes |
| Doyle et al. [73] | USA | 18–78 years | IBS (100) | Colon | Anti-kit | 5 | Area of highest density | Yes |
| Coeffier et al. [74] | France | Mean 44.6 years | IBS (25) | Descending colon | CD117 | 10 | Not stated | Yes |
| Walker et al. [75] | Sweden | Mean 53 years | FD (51) and IBS (41) | Duodenum | CD117 | 5 | Not stated | Yes |
| Taki et al. [76] | Japan | Mean 53 years | FD (35) | Duodenum | CD117 | $\geq 3$ | Representative | Yes |
| Lee et al. [77] | Korea | Mean 36 years | FD (51) | Duodenum | c-KIT | 5 | Hot spots | No |
| Wauters et al. [78] | Belgium | 18–64 years | FD (45) | Duodenum | c-kit | 3 | Not stated | Yes |

**Table 3.** Summary of adult studies assessing mast cell density which also assessed mast cell activation in functional abdominal pain disorders.

| Author | Country | Age Group | Population (N) | Mucosal Sites | Mast Cells ID Method | Number of Microscopic Fields Assessed | Field Selection | Density Different from Controls | Activation Different from Controls |
|---|---|---|---|---|---|---|---|---|---|
| Park et al. [10] | Korea | Mean 48 years | IBS-D (14) | Cecum and rectum | Toluidine blue | Up to 20 | Not stated | Yes | Yes |
| Liu et al. [11] | China | 22–40 years | IBS-D (42) | Rectosigmoid junction | Toluidine blue | 5 | Random | No | Yes |
| Xu et al. [12] | China | 18–49 years | IBS-D (38) | Rectosigmoid junction | Toluidine blue | 5 | Random | Yes | No |
| Yuan et al. [13] | China | Mean 45–47 years | FD (48) | Duodenum | Toluidine blue | Not stated | Not stated | Yes | Yes |
| Yuan et al. [14] | China | Mean 45–47 years | FD (48) | Duodenum | Toluidine blue | Not stated | Not stated | Yes | Yes |
| Wang et al. [15] | China | Mean 46 years | FD (141) | Duodenum | Toluidine blue | 4-6 random sites, then 5 | Random | Yes | Yes |
| Foley et al. [16] | England | Mean 42 years | IBS-D (20) | Duodenum | Tryptase | Not stated | Not stated | Yes | Yes |
| Lee et al. [24] | Korea | 24–66 years | IBS-D (16) | Rectum | Tryptase | 5 | Not stated | No | Yes |
| Barbara et al. [17] | Italy | 22–75 years | IBS (44) | Descending colon | Tryptase | Area occupied | Random | Yes | Yes |
| Balestra et al. [25] | Italy | 21–70 years | IBS (37) | Descending colon | Tryptase | % of LP occupied | Random | Yes | Yes |
| Han et al. [26] | China | 18–59 years | PI-IBS (23) | Left colon | Tryptase | ≥8 | Not stated | Yes-area; No-number | Yes |
| Cremon et al. [27] | Italy, Spain, France, Croatia, and Bosnia and Herzegovina | Mean 37–40 years | IBS (54) | Proximal descending colon | Tryptase | Not stated | Not stated | Yes | Not reported |
| Bednarska et al. [28] | Sweden | 19–55 years | IBS (32) | 30-40 cm from anal verge | Tryptase | Not stated | Not stated | Yes | Yes |
| Buhner et al. [29] | Italy | 27–68 years | IBS (11) | Proximal descending colon | Tryptase | Not stated | Not stated | Yes | Yes |

**Table 3.** *Cont.*

| Author | Country | Age Group | Population (N) | Mucosal Sites | Mast Cells ID Method | Number of Microscopic Fields Assessed | Field Selection | Density Different from Controls | Activation Different from Controls |
|---|---|---|---|---|---|---|---|---|---|
| Barbara et al. [30] | Italy | 19–70 years | IBS (29) | Proximal descending colon | Tryptase | Not stated | Not stated | Yes | Yes |
| Li et al. [20] | China | 17–65 years | FD (65) | Antrum | Tryptase | 10 | Not stated | Yes | Yes |
| Vanheel et al. [79] | Belgium | 23–43 years | FD (24) | Duodenum | Tryptase | $\geq 7$ | Representative | Yes | No |
| Du et al. [19] | China | Mean 48 years | FD (96) | Duodenum | Tryptase | 5 | Random | Not reported | No |
| Cremon et al. [31] | Italy | 22–56 years | IBS (25) | Descending colon | Tryptase | Area occupied | Random | Yes | Yes |
| Klooker et al. [32] | The Netherlands | 19–65 years | IBS (29) | Descending and rectum | Tryptase or CD117 | 18 | Not stated | Yes- decreased | Yes- decreased |
| Martinez et al. [21] | Spain | 18–60 years | IBS-D (45) | Jejunum | CD117 | Not stated | Not stated | Yes | Yes |
| Vivinus-Nébot et al. [33] | France | 42–58 years | IBS (34) | Cecum | CD117 | 3 | Not stated | Yes | Yes |
| Lobo et al. [18] | Spain | 18–65 years | IBS-D (43) | Jejunum | CD117 | 10 | Not stated | No | Yes |
| Guilarte et al. [22] | Spain | 21–56 years | D-IBS (20) | Jejunum | CD117 | 8 | Not stated | Yes | Yes |
| Martinez et al. [23] | Spain | 18–59 years | IBS-D (25) | Jejunum | CD117 | Not stated | Not stated | Yes | Yes |

Only 2 pediatric studies compared patients to a control group and both studies demonstrated increased mast cells in the study group [36,39]. Neither assessed activation. Results comparing adult patients to controls are shown in Tables 2 and 3. Overall, density in comparison to controls was reported in 57 studies. Of these, increased density was reported in 45 studies (79%), no difference in 10 studies (18%), and decreased density in 2 studies (4%). One study reported density as both cell count and area occupied by mast cells demonstrating decreased cell counts and increased area occupied in patients with IBS [26]. Increased mast cells in association with an FAPD was found in 26/33 (79%) of studies staining for tryptase, 12/15 (80%) of studies staining for CD117, and 8/10 (80%) of studies utilizing histochemical staining.

Overall activation in comparison to controls was reported in 24 studies (Table 3). Of these, increased activation was reported in 20 studies (83%), decreased activation in one study (4%), and no difference in 3 studies (13%). Twenty-three of these studies reported both density and activation as compared to controls. In 5 studies, there was a discrepancy between density and activation findings. Three studies showed increased activation with no difference in density and two showed increased density with no difference in activation. One study showed increased density by area occupied but not cell density and activation was increased. Seven studies utilized multiple methods to assess activation and findings were concordant between methods in 6 studies [17,18,20,26,28–30]. In the remaining study, supernatant tryptase was increased but degranulation did not differ from controls [28]. Overall activation was assessed by supernatant tryptase in 12 studies, degranulation in 10 studies, tissue tryptase expression or protein analysis in 5 studies, supernatant histamine in 6 studies, assessment of in vitro nerve stimulation in 3 studies, and luminal tryptase in 2 studies.

## 4. Discussion

A variety of techniques are available for assessing mast cell density in FAPDs and these have been applied with considerable variability. Immunohistochemistry (IHC) staining appears to be the preferred method for identifying mast cells to assess density, with anti-tryptase or anti-CD117 antibodies utilized in 73% of pediatric studies and 83% of adult studies. Histochemical stains for mast cells are known to be less sensitive in identifying mast cells in gastrointestinal mucosa fixed with formalin [80,81]. Not surprisingly, there appears to be a preference for IHC stains. However, these IHC stains are not without limitations and introduce other variability into the literature. For example, 20–30% of tryptase-positive cells in the stomach and colon fail to stain for CD117, creating a challenge in assimilating data obtained utilizing the two different methods [82]. CD117 is present in immature mast cells and a large proportion of mast cells in the stomach and colon do not stain with anti-CD117 [83]. While anti-tryptase appears to identify more mast cells and is the most commonly utilized method, tryptase is expressed late in mast cell maturation and will not identify those mast cells expressing only chymase which are present in the stomach, small bowel, and colon [81–83]. This latter limitation can be overcome by also staining for chymase [81]. Variability is also introduced by the selection of microscopic fields to be assessed. While the process for field selection is often not stated, when reported, it varies from random to "most representative" to most involved. The rationale for most involved is that density is often patchy. Most studies assess mast cells per area, either per high power field (hpf) or per mm$^2$. The actual area of a hpf varies between microscopes. Others assess the percentage of area occupied by mast cells utilizing digital imaging. While some studies have found a high correlation between manual cell counts and measurement of the percentage of area occupied by mast cells, another study found discordant results between manual counts and area occupied in comparison to healthy controls [26,52].

There are also a variety of methods for assessing mast cell activation. This may be of particular importance as most biologic functions of mast cells are the result of the release of specific mediators generally acting in a concentration-dependent fashion [84]. There appear to be 3 commonly used approaches for assessing activation: (1) assessing

degranulation at a cellular level using light microscopy or at both a cellular and sub-cellular level using electron microscopy, (2) assessing tissue or luminal mast cell mediators, most commonly tryptase and histamine, and (3) functional studies using mucosal supernatants to stimulate enteric nerves in vitro. Regardless of the method, increased activation relative to healthy controls was demonstrated in over 80% of studies. In 22% of studies where activation was assessed by any method, there was a discrepancy between density and activation comparisons to controls. In 2 studies, density alone was increased and in 3 studies, activation alone was increased. Density and activation measurements are not perfectly aligned and both may be needed to get a full picture of mast cell involvement. When multiple methods were utilized to assess activation, these were concordant with each other in 6 of the 7 studies.

While mast cells produce a wide variety of cytokines, chemokines, and other mediators, previous studies have primarily, but not exclusively, focused on tryptase and histamine. This appears justified as both have been implicated in mast cell interactions with sensory nerves. In a series of experiments, Wouters and colleagues established a role for histamine (via H1 receptors) in upregulating TRPV1 which has been highly implicated in visceral hyperalgesia, a central process in FAPDs [9]. Studies have nearly universally found increased tryptase in tissue, tissue supernatants, and luminal fluid. Studies have also demonstrated that supernatants from IBS mucosal biopsies can stimulate submucosal sensory nerves which correlate with serotonin, histamine, and tryptase concentrations and which could be inhibited by H1 receptor antagonists and serine protease inactivation [29,30]. An effect on myenteric nerves is reported to be independent of histamine and serine protease [25]. Mast cell mediators appear to have differential effects on submucosal and myenteric nerves [85]. Methods utilizing sensory nerves are likely a better model for assessing mast cell effects in FAPDs and possibly more representative of effects on pain transmission and visceral hyperalgesia.

There is some evidence to support mast cells as therapeutic targets in IBS and FD utilizing medications to inhibit mediator release (e.g., mast cell stabilizers such as cromolyn or ketotifen or anti-siglec 8) or to inhibit mast cell mediators once released (e.g., histamine or cysteinyl leukotriene inhibitors) [18,32,86–91]. One study utilizing cromolyn and one utilizing ketotifen demonstrated evidence of decreased mast cell activation [18,32]. However, no studies have evaluated whether any measures of mast cell density or activation are predictive of response to any of these medications.

## 5. Conclusions

A wide variety of methods have been employed to assess mast cell density and activation in patients with FAPDs with no well-established consensus and not enough data to recommend a specific approach. Given that differences in mast cell density and activation between patients with FAPDs and healthy controls are demonstrated in a strong majority of studies, regardless of methodology, this variability may make a more compelling case for mast cells in the pathophysiology of FAPDs in general. Whether this evidence is strong enough to warrant empiric treatment aimed at mast cell stabilization or mediators is up to clinical interpretation. Ideally though, the methodology providing physiologic information would be translated to a standard methodology providing clinical information with the development of criteria establishing abnormal density and/or activation, and more importantly, predicting response to treatment.

**Author Contributions:** Conceptualization, H.F., M.S., V.S. and J.V.S.; methodology, H.F., M.S., V.S., J.V.S. and C.A.F.; data curation, H.F. and C.A.F.; writing—original draft preparation, H.F. and C.A.F.; writing—review and editing, H.F., M.S., V.S., J.V.S. and C.A.F. All authors have read and agreed to the published version of the manuscript.

**Funding:** This research received no external funding.

**Institutional Review Board Statement:** Not applicable.

**Conflicts of Interest:** The authors declare no conflict of interest.

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
