# Peer review of "A Survey of Methodologies for Assessing Mast Cell Density and Activation in Patients with Functional Abdominal Pain Disorders"

_gastrointestdisord, doi:10.3390/gidisord3040016_

Round 1

Reviewer 1 Report

The authors evaluate methods utilized in assesing mast cell involvement in functional abdominal pain disorders (FAPDs), specifically to describe variability in methods utilized to assess both mast cell density and activation and determine if a consensus exists. 11 studies in children and 59 in adults were analysed. It seems that the presentation of the results from all 70 articles in the tables is not very readable. I propose to group the results of the work taking into account, for example, the place of taking the samples or the methodology of histopathological examinations performed. Table 3 also needs to be amended above.  The conclusions of this work are very practical and important. The authors conclude that the current methodology providing physiologic information needs to be translated to a standard methodology providing clinical information with development of criteria establishing abnormal density and/or activation, and more importantly, predicting treatment response.

Author Response

We thank the reviewer for their recommendations and have done the following to address their concerns:

  1. We defined IBS and FD with appropriate references on lines 33-36.
  2. We added references regarding the methods on lines 59-62.
  3. The period of time for the literature search was added on line 73. 
  4. The only exclusion criteria was added on lines 79-80.
  5. As this was a methodologic review (as opposed to a systematic review), the editor asked us to format it as a review eliminating the Methods and Results headings. Strengths and weaknesses did not seem to fit under the review format. We would be happy to comment separately on the strengths and weaknesses of the existing literature but would ask direction from the editor. 

Reviewer 2 Report

Introductory part: the authors should better define IBD and FD, according to the latest guides (and cite appropriate studies). 

Raws 57-60: I recommend to cite some articles that mention these methods. 

Materials and methods: the period of time for the search of the literature should be specified. 

The exclusion criteria for the studies should be also clearly stated. 

Discussion section: the strengths and limitations of this review study should be stated.  

Author Response

We thank the reviewer for their recommendations and have done the following to address their concerns which we believe increases the readability and utility of the manuscript:

We divided the original Table 2 into two tables: A new Table 2 for adult studies where activation was not assessed and Table 3 for adult studies where activation was also assessed. Within each of these table, we re-ordered the studies such that all studies utilizing histochemical methods were together, all IHC stains for tryptase were together, and all IHC stains for CD117 (c-kit) were together. In addition, this allowed the useful information from the original Table 3 to be incorporated into the new tables eliminating the original Table 3. We appreciate the recommendation and feel this makes the information much more usable. We would be happy to make further alterations as recommended.  

Round 2

Reviewer 2 Report

The authors made all the recommended changes. The value of the manuscript increased and I recommend publication.